# Familial Dilated Cardiomyopathy: A Novel MED9 Short Isoform Identification

**DOI:** 10.3390/ijms25053057

**Published:** 2024-03-06

**Authors:** Monica Franzese, Mario Zanfardino, Andrea Soricelli, Annapaola Coppola, Ciro Maiello, Marco Salvatore, Concetta Schiano, Claudio Napoli

**Affiliations:** 1IRCCS SYNLAB SDN, 80143 Naples, Italy; monica.franzese@synlab.it (M.F.); mario.zanfardino@synlab.it (M.Z.); andrea.soricelli@uniparthenope.it (A.S.); direzionescientifica.irccssdn@synlab.it (M.S.); claudio.napoli@unicampania.it (C.N.); 2Department of Exercise and Wellness Sciences, University of Naples Parthenope, 80133 Naples, Italy; 3Department of Advanced Medical and Surgical Sciences (DAMSS), University of Campania Luigi Vanvitelli, 81100 Naples, Italy; annapaola.coppola@unicampania.it; 4Department of Cardiothoracic Science, U.O.S.D. of Heart Transplantation, Monaldi Hospital, 80131 Naples, Italy; ciromaiello64@gmail.com; 5Clinical Department of Internal Medicine and Specialistic Units, Division of Clinical Immunology, Immunohematology, Transfusion Medicine and Transplant Immunology (SIMT), Azienda Universitaria Policlinico (AOU), 80131 Naples, Italy

**Keywords:** RNA sequencing, heart failure, dilated cardiomyopathy, precision medicine, atherosclerosis, mediator complex

## Abstract

Familial dilated cardiomyopathy (DCM) is among the leading indications for heart transplantation. DCM alters the transcriptomic profile. The alteration or activation/silencing of physiologically operating transcripts may explain the onset and progression of this pathological state. The mediator complex (MED) plays a fundamental role in the transcription process. The aim of this study is to investigate the MED subunits, which are altered in DCM, to identify target crossroads genes. RNA sequencing allowed us to identify specific MED subunits that are altered during familial DCM, transforming into human myocardial samples. N = 13 MED subunits were upregulated and n = 7 downregulated. MED9 alone was significantly reduced in patients compared to healthy subjects (HS) (FC = −1.257; *p* < 0.05). Interestingly, we found a short MED9 isoform (MED9s) (ENSG00000141026.6), which was upregulated when compared to the full-transcript isoform (MED9f). Motif identification analysis yielded several significant matches (*p* < 0.05), such as GATA4, which is downregulated in CHD. Moreover, although the protein–protein interaction network showed FOG2/ZFPM2, FOS and ID2 proteins to be the key interacting partners of GATA4, only FOG2/ZFPM2 overexpression showed an interaction score of “high confidence” ≥ 0.84. A significant change in the MED was observed during HF. For the first time, the MED9 subunit was significantly reduced between familial DCM and HS (*p* < 0.05), showing an increased MED9s isoform in DCM patients with respect to its full-length transcript. MED9 and GATA4 shared the same sequence motif and were involved in a network with FOG2/ZFPM2, FOS, and ID2, proteins already implicated in cardiac development.

## 1. Introduction

Dilated cardiomyopathy (DCM) impairs the heart’s ability to efficiently pump blood around the body [1]. DCM primarily affects the left ventricle, causing an enlargement, which is in turn associated with systolic heart failure (HF) or low ejection fraction (EF) [1]. Although DCM can be asymptomatic, if left untreated, it has the potential to lead over time to HF, a syndrome characterized by pulmonary, abdominal, upper, and lower limb congestion, mitral and/or tricuspid valve insufficiency secondary to ventricular dilation, embolisms, and arrhythmias, all issues capable of causing sudden death [2]. To improve symptoms and increase survival, when the cause of DCM becomes known, it should either be removed or corrected [3]. During HF and depending on the severity of the disease, several treatment options are available. In definite clinical situations, a large proportion of patients suffering from HF, undergoing optimized pharmacological therapy and faced with reduced left ventricular function, may resort to the implantation of a biventricular pacemaker (PM) or an automatic defibrillator (ICD) following precise risk-stratification criteria specified to the individual according to current guidelines [https://academic.oup.com/eurheartj/article/44/37/3503/7246608?login=true#427681730, accessed on 27 February 2024]. When implanting a cardiac resynchronization therapy (CRT) pacemaker, or when ICD is no longer sufficient, there is the option of using a left ventricular assist device (LVAD). This latter approach is chosen in cases of very advanced HF, when it is necessary to provide the patient with the best possible conditions to accompany the heart transplant (HTx) and therefore enable the subject to overcome the waiting period [3].

Over the years, there has been increasing evidence that DCM is a familial or genetic disease. To improve DCM care, it is necessary to obtain a better understanding of the etiological basis of the disease, perform appropriate risk stratification, and develop new therapies. DCM manifests diverse inheritance patterns, illuminating its genetic underpinnings. Predominantly, autosomal dominant transmission prevails as the primary mode, characterized by the heterozygous inheritance of pathogenic alleles, conferring susceptibility upon carriers. This genetic paradigm entails a 50% likelihood of vertical transmission from an affected parent, fostering familial aggregation. Concurrently, X-linked inheritance delineates an alternate genetic trajectory, mediated by genes harbored on the X chromosome, thereby accentuating sex-dependent penetrance. In X-linked DCM, males are typically affected, while females may exhibit milder symptoms or act as carriers. Moreover, mitochondrial inheritance underscores the matrilineal propagation of DCM-associated mitochondrial DNA variants, engendering the susceptibility of progeny. Elucidating the nuanced inheritance modalities of DCM imparts crucial insights into its molecular etiology, informing tailored genetic counselling paradigms and therapeutic interventions [4]. Specifically, it is known that familial DCM is caused by defective genes that affect the function of the heart muscle. Several familial DCM genes are currently known, whereas others are still under investigation; therefore, the knowledge of possible alterations in the transcriptional profile can contribute to explaining the onset and/or progression of the disease [2,3]. Notably, mutations in several genes have been implicated in DCM pathogenesis, shedding light on its heterogeneous etiology. Among these, mutations in the “titin” gene have garnered significant attention, with studies linking titin mutations to DCM development and progression. Furthermore, mutations in the LMNA gene, desmoplakin gene, and dystrophin gene have also emerged as key contributors to DCM pathology [5,6,7,8]. These genetic variants act on structural and functional components of cardiac muscle, culminating in myocardial dysfunction characteristic of DCM. Incorporating these specific gene mutations into our understanding of DCM underscores the intricate interplay between genetic factors and disease phenotype. Alterations in the regulation of gene expression are often associated with different CVDs, including HF, arrhythmias, and coronary syndromes [9,10,11,12,13]. MED plays a key role in the transcription mechanism activation, being a multi-protein complex essential in both the initial stages of RNA polymerase II recruitment and the subsequent stages of processing and process termination. Specifically, MED transduces signals from activation factors, which are assembled into promoters to form the pre-initiation complex (PIC) and initiate transcription. Recent structural and functional biology studies have revealed new details about interactions between MED and transcription regulatory regions [10,14,15]. Our assumption is that only some subunits of the MED can activate, organize themselves, and assume a conformation suitable for the regulation of specific signal pathways downstream. Conversely, other subunits are capable of being deactivated, encouraging future studies for the identification of the disease-specific MED “core”. The detection of novel molecular targets underlying the onset of DCM may reduce morbidity and mortality by allowing the design of personalized therapeutic approaches.

In our study, we have performed a transcriptome analysis of HTx patients in order to identify target transcripts, which could recognize pathological subjects from healthy ones significantly. Focusing our attention on the alterations occurring in the “mediator” protein complex (MED), which plays a fundamental role in the gene transcription mechanism in cooperation with RNA polymerase II (RNA Pol II), our specific aim was to identify the subunits of MED that primarily assemble and initiate the transcription of genes, which in turn code for functionally altered proteins or factors during DCM.

To identify specific crucial DCM targets, we applied an NGS approach, specifically RNA-seq, on myocardial tissue samples taken from both DCM patients and healthy donors during organ transplant surgery. Then, utilizing a traditional analytical approach, we evaluated the difference in gene expression [16]. The transcriptome analysis enables us to identify genes with notable differences in their expression levels, which characterize the biomolecular mechanisms underlying a particular condition. It is urgent to find an adequate treatment, enabling the identification of the underlying pathophysiology that can drive future treatments aiming to repair or replace the existing gene mutation or target the specific epigenetic and/or inflammatory drivers of genetic or acquired DCM. Our goal is to identify the early cascades of deregulated signaling pathways. This would then allow us to acquire a more complete understanding of the phenotype of interest and could help to uncover new target genes involved. Understanding the right signaling pathways could therefore facilitate the identification of new therapeutic strategies and provide doctors with an interesting tool capable of distinguishing a patient with a functionally damaged heart from a healthy subject.

## 2. Methods

### 2.1. Study Design and Baseline Patient Characteristics

The study was conducted according to the protocols approved by the Ethics Committee of the “Monaldi” Hospital and the “Federico II” Hospital (79/18, 11 May 2018) and in compliance with the principles outlined in the Declaration of Helsinki. Specifically, the study included n = 50 subjects, of which n = 11 were organ donors who died due to accidental death and acted as healthy controls. HF patients (n = 39) were divided into two groups: DCM (n = 11) and no DCM (n = 28). Among the 2 groups, only the DCM group which underwent HTx. Among the 2 groups, only the DCM group that reporting a reduced ejection fraction (HFrEF), underwent HTx and processed via NGS. This group included patients with determinate familial diagnosis. Familial DCM was diagnosed based on patient medical history and cardiac imaging, such as transthoracic echocardiography and coronary angiography. Laboratory tests, including assessments of hemoglobin (13.54 ± 1.9 mg/mL), hematocrit (39.30 ± 4.2%), hsCRP (4.8 ± 0.8 mg/l), total cholesterol (143.30 ± 48.6 mg/dl), and hsTnT (29.9 ± 9.2 pg/mL), were obtained for the DCM group. The subjects were not age-matched, with an average age of 50 (49.4 ± 16.1) years for the patients and 31 (30.6 ± 13.1) years for heart donor (*p* < 0.01). Some 65% of subject were men. Left ventricular end-diastolic diameter and left ventricular end-systolic diameter were considered as echocardiographic parameters, reporting measurements of 7.05 ± 0.72 and 6.13 ± 0.81 mm, respectively. Control myocardial samples were acquired from the apical left atrial and ventricular tissues of donors, during autopsy, and from non-failed heart donors who died from non-cardiac accidental causes. Myocardial tissue samples were collected and immediately frozen in liquid nitrogen at the time of surgery and were subsequently used in the RNA-seq and qRT-PCR experiments, respectively.

Informed consent was obtained from all individual participants included in the study.

### 2.2. RNA Extraction, Sequencing, and Data Analysis

Total RNA was extracted from all tissues of donor and recipient hearts. The quality and quantity of RNA were subsequently assessed using the Nanodrop apparatus [17,18]. The cDNA libraries were prepared using the Illumina TruSeq RNA Sample Preparation Kit and sequenced on the Illumina HiSeq2000 (Illumina, San Diego, CA, USA) [16]. For the analysis of the whole transcriptome, high-quality reads were mapped onto the human reference genome. RNA-seq data were obtained from a total of n = 22 subjects (n = 11 DCM and n = 11 HS). Based on statistical filtering of approximately 9.000 genes with a Bonferroni corrected *p*-value < 0.05, we selected 1187 differential expressed genes (DEGs). Raw sequencing data (fastq files) of biological samples were submitted to the NCBI BioProject database with login ID PRJNA667310 and GSE71613 (https://www.ncbi.nlm.nih.gov/search/all/?term=PRJNA667310, accessed on 27 February 2024).

### 2.3. Bioinformatic Analysis

We performed quality control on raw reads using FastQC (v0.11.9) (http://www.bioinformatics.babraham.ac.uk/projects/fastqc/, accessed on 27 February 2024). High-quality reads were mapped to the human reference genome (GRCh38) using HISAT2 (v2.1.0) [19]. The resulting sam files were converted to a bam format using Samtools (v1.9) [20]. A list of n = 33 mediator subunits was extracted from a general-purpose read summarization analysis (featureCounts function from Rsubread R package v2.0.1) [21,22] and the counts were compared in healthy/DCM conditions via differential expression analysis (edgeR, R package v3.38.4). A selected mediator, which was significantly differentially expressed in the two conditions, was used as input for motif identification analysis, starting from a cDNA sequence (MEME suite v5.5.3) [23]. From the results of motif analysis, we selected a gene of particular interest in CHD and, using this gene together with the list of all DEGs, we built a protein–protein interaction (PPI) network using STRING v11.0 [24]. Additionally, a gene ontology (GO) analysis was performed via the use of a STRING tool on the proteins taken from this network.

### 2.4. Data Validation via qRT-PCR

Quantitative reverse transcriptase PCR (qRT-PCR) was carried out to validate the obtained data. A small quantity of total RNA (250 ng for each sample) was reverse-transcribed to complementary DNA (cDNA) using the SuperScript IV First-Strand cDNA Synthesis Reaction (Invitrogen, Carlsbad, CA, USA) following the manufacturer’s protocol. The BLAST program and in silico PCR analysis were utilized to evaluate the specificity of each oligonucleotide pair used [25]. The Genome Browser tool (https://genome.ucsc.edu, accessed on 27 February 2024) was used to select oligonucleotides used in the qRT-PCR analyses. Appendix A reports the primer sequences used. Calculated through the 2^−ΔΔCt^ method, relative expression levels of the target genes normalized with RPS18 were reported as fold change (FC) [25].

### 2.5. Statistical Analysis

The Shapiro–Wilk normality test was used to determine the normality of data. Results are shown as the mean ± SE. Statistical analysis was performed via Student’s t-test for paired data. Continuous variables were expressed as mean ± standard error (SE). Data were tested for normality through the Shapiro-Wilk test. The unpaired Student’s t-test or Mann–Whitney U test was used as required for comparison between two groups. A significant value * was expressed with *p* < 0.05 vs. HS, ** with *p* < 0.01 vs. HS.

## 3. Results

### 3.1. Study Population

The screening of hospital records identified newly diagnosed familial DCM patients reporting HFrEF. These patients were categorized according to the New York Heart Association (NYHA).

### 3.2. MED and DCM

Using RNA-seq technology, we studied the changes occurring in the cardiac transcriptome of DCM patients undergoing organ transplantation. We performed bioinformatic analysis, revealing the approximately n = 8500 DEGs indicated as protein-coding genes (Appendix A). Selective analysis enabled the observation that the DEGs included several MED subunits. All MEDs differentially expressed in DCM vs. HS were reported in Appendix A. Specifically, n = 7 subunits belonged to the head (MED6, MED8, MED17, MED20, MED22, MED28, and MED29); n = 6 to the middle (MED1, MED4, MED7, MED9, MED14, and MED21); and n = 3 to the tail module (MED15, MED23, and MED24). Additional, n = 4 constituted the kinase module (MED12, MED13/MED13L, and CDK8/19), as reported in Appendix A. We generated a MED heatmap, which allowed us to combine the MED DEGs into DCM and HS groups, as shown in Figure 1A. The quantitative analysis showed that the 35% (n = 7) (MED4, MED6, MED7, MED8, MED9, MED20, MED21) of the MED was downregulated, while 65% (n = 13) showed upregulation, including values between logFC −0.33 and +0.35. In particular, MED9 was the most downregulated subunit, whereas MED12 was the most upregulated variant. MED9 was also the only one to show a statistically significant difference in the expression trend between DCM and HS groups (*p* < 0.05), as reported in the Appendix A and Figure 1B. Interestingly, when we performed qRT-PCR, we observed that the MED9 full-length transcript (MED9f) showed a reduced expression level between DCM and HS groups, confirming the trend found into transcriptome sequencing. Conversely, another transcript isoform, which we named MED9 short (MED9s) and which is already annotated in Genome Browser tool (https://genome.ucsc.edu/, accessed on 27 February 2024), was significantly increased in DCM compared to HS (*p* = 0.027) (Figure 1C).

### 3.3. MED9 Motif Enrichment Analysis

Since MED9 produced significantly different (*p* < 0.05) results in the two analyzed conditions (DCM vs. HS), we performed motif enrichment analysis (MEA). Starting from the cDNA sequence, this allowed for the unbiased identification of TFs that exhibit large changes in chromatin accessibility at sites containing their DNA-binding motifs. MEA produced 30 significant matches (*p* < 0.05), as reported in Appendix A. From these results, we selected GATA-binding protein 4 (GATA4), a zinc finger transcription factor that showed a prominent involvement in DCM pathology (Figure 2). The protein product of GATA4 gene is essential for normal cardiac development and homeostasis in both mice and humans, and its mutations have been reported in human heart defects [26].

### 3.4. PPI Network Analysis and Gene Ontology (GO)

Due to the emergence of MEA GATA4 as a gene of particular interest in heart defects, we performed a protein–protein interaction (PPI) network analysis using GATA4 and DEGs genes from differential expression analysis (Figure 3A). The data output provided a network with n = 1187 nodes and n = 4802 edges when filtering using a “medium confidence” score (>0.4, average score = 0.79). GATA4 interacts with 9 proteins with a combined score average of 0.31 (derived from experimentally determined interaction alone). Specifically, Zinc Finger Protein, FOG Family Member 2 (ZFPM2/FOG2), a transcriptional repressor that binds to members of the GATA family of transcription factors, produced an experimentally determined interaction with GATA4 (“high confidence” score: 0.87). Moreover, we also selected FOS Proto-Oncogene, AP-1 Transcription Factor Subunit (FOS), which is the highest differentially expressed gene among the protein interacting with GATA4. Finally, we focused on the inhibitor of differentiation/DNA binding protein 2 (ID2) that shows an interesting role in cardiac morphogenesis and development, as established via GO analysis (Figure 3B) [27]. Moreover, it cooperates with GATA4 transcription factors for the development of the cardiac conduction system [28]. The four selected proteins (GATA4, FOG2/ZFPM2, FOS, ID2) show an interesting GO class of membership with FDR < 0.05 (Figure 3B).

### 3.5. Hub DCM-Related Gene Validation

First, we validated all isoforms of the MED9 gene. Furthermore, since motif identification and PPI analysis highlighted several MED9-coniugated targets, we validated the list of potential genes that could be involved, whether directly or indirectly, in DCM onset and progression (Appendix A). We performed qRT-PCR for the following hub genes: GATA4, FOG2/ZFPM2, FOS, and ID2. GATA4 is a critical regulator of cardiac differentiation; FOG2/ZFPM2 is essential for heart morphogenesis and coronary development; FOS is significantly activated in dilated cardiomyopathy. Finally, we also proceeded to validate ID2 because it cooperates with GATA4 transcription factors for the development of the cardiac conduction system [28]. Interestingly, the selected target genes showed expression trends similar to those visible in the literature [26,27,29]. At the validation step, we found that GATA4 showed significant downregulation (FC 0.43 ± 0.28) (*p* < 0.05), whereas FOG2/ZFPM2 was significantly upregulated (FC 3.57 ± 1.63) (*p* < 0.05) in DCM patients. Finally, FOS and ID2 were not significantly increased (FC times, 2.39 ± 1.30, and 1.86 ± 1.34 times, respectively) in DCM vs. HS (Figure 3C). Specifically, the relative expression was calculated as fold change (FC) between DCM and HS subjects and assigned a reference value = 1 at the control group (HS).

## 4. Discussion

In this retrospective, single-center study of n = 11 newly diagnosed DCM patients who underwent HTx, we reported a significant deregulation of MED during HF. For the first time, we demonstrated that the MED9 subunit showed a significantly reduced expression level between DCM patients and healthy controls (*p* < 0.05) (Appendix A). Combining computational and experimental approaches, we became the first to identify a novel alternative transcript for MED9, called MED9s. In contrast to the full-length isoform (MED9f), this is significantly overexpressed in DCM compared to HS. The MED complex is one of the key players in gene transcription machinery; therefore, the identification of MED multiprotein deregulation could be extremely important to uncover the specific altered target genes and promote future diagnostic/prognostic applications. To date, there are still few studies evaluating midbody subunits during CVDs, while studies investigating the role of MEDs in CHD are entirely absent from the literature. In addition, it is difficult to study early pathogenetic events related to the development of DCM in people. This is not only due to the difficulty of conducting an in vivo biopsy before HTx, but also due to the lack of methods enabling an early differential diagnosis [30,31]. The literature reports that only MED12, MED13, MED13L, MED15, MED23, and CDK8 have an important involvement in cardiac development and cardiac malformations, while some subunits, such as MED1, MED14 and CDK8, are indirectly involved in adipogenesis and lipid metabolism [9,32]. In this regard, most of studies derive information from in vitro observations and animal models, and there is only human evidence for MED12, MED13, MED13L and MED15 [9,32]. For this reason, our study focused on the composition of MED during advanced DCM. Although the cause of more than 80% of CAD remains unexplained, the literature reports that a subset of cases is generated by Mendelian disorders, specific chromosomal abnormalities, and copy-number variants (CNVs). The identification of CAD loci that were altered by CNVs results is very important, offering a superb opportunity to elucidate CAD pathogenesis. Using two independent strategies, namely, high-density array SNP genotyping (Illumina Omni-1.0 and 2.5 M) and whole-exome sequencing (WES), it was possible to detect small de novo CNVs, including genes for some MED subunits (MED9 and MED15), in a family study of sporadic cases of CAD with conotruncal outflow tract defects, heterotaxy syndrome, and left ventricle defects [33]. Specifically, these single CNVs have been observed to include a set of genes that interact with established CAD proteins, such as GATA4 [33]. Recently, MED9, MED16, MED18, and CDK8 were identified as playing specific roles in molecular mechanisms, with connections to different stress-signaling pathways [34]. Crystallographic studies show that the middle module contains several flexible domains, like the “Head” module. The MED9/MED4 heterodimer adopts a similar fold to MED7/MED21, probably harboring a similar hinge [35]. Different conformations of both modules can be expected when they are trapped in higher-order complexes, such as an intact central MED, under physiological conditions, or altered Pol II complexes, during pathological events. According to some observations made in plants, a study also indicated a reduction in MED16, which could be related to a co-repression function during stress [34]. These results underscore the crucial role of these MEDs as poorly understood but important players in the transcriptional stress response. In addition, our results have identified MEDs target genes and their related molecular mechanisms as a prelude to further biochemical characterization. The MED9s isoform had already been identified and characterized previously [36]. MED9s was transcribed into endothelial progenitor cells (EPCs) and was absent in the differentiated endothelial cells (ECs) [36]. Therefore, we hypothesized that cardiac cells might be less differentiated in DCM patients, corroborating MED9 transcription. The motif identification analysis yielded several significant matches with MED9 (*p* < 0.05) (see the Appendix A) such as GATA4, a zinc finger-containing, DNA-binding transcription factor essential for normal cardiac development and homeostasis both in mice and humans. Moreover, mutations in GATA4 gene have been reported in human heart defects [26]. Interestingly, Zhu L. et al. reported that GATA4 directly binds to many mRNAs through defined RNA motifs in a sequence-specific manner. In vitro splicing assays indicated that GATA4 regulates alternative splicing through direct RNA binding, resulting in functionally distinct protein products [26]. In our study, we hypothesized that GATA4, interacting with spliceosome complex members, could induce the expression of MED9 alternative isoform (MED9s), thus regulating cell cardiac-specific alternative splicing via sequence-specific interactions with RNA, in addition to features previously reported [26]. Finally, the PPI network also showed that some proteins, such as FOG2/ZFPM2, FOS and ID2, could be regulated in DCM patients. In agreement with the literature, our results confirmed that FOG2/ZFPM2 was significantly upregulated, whereas FOS and ID2 were non-significantly increased in the DCM group vs. the HS group. In conclusion, GATA4-driven alternative splicing of MED9 could have functional consequences in DCM, suggesting a novel role for MED9 in the cardiac proteome under physiological conditions.

## 5. Conclusions

Our study provides insights into the molecular mechanisms underlying DCM via RNA-seq analysis of cardiac transcriptomes in DCM patients undergoing organ transplantation. The identification of several differentially expressed genes (DEGs), including several subunits of the MED, highlights the dysregulation of transcriptional machinery in DCM pathophysiology. Notably, MED9 emerges as a pivotal mediator, exhibiting significant downregulation in DCM patients compared to healthy subjects, implicating its potential role in disease progression. MEA further underscores the involvement of GATA4, a zinc finger transcription factor, in DCM pathology, corroborating previous findings regarding its crucial role in cardiac development and homeostasis. PPI network analysis reveals intricate relationships between GATA4 and key regulatory proteins, such as FOG2/ZFPM2, FOS, and ID2, shedding light on their collective impact on cardiac function. The validation of hub genes, including GATA4 and FOG2/ZFPM2, through quantitative PCR reaffirms their dysregulation in DCM, implicating them as potential therapeutic targets. Our findings underscore the complex interplay between transcriptional regulators in DCM pathogenesis, paving the way for targeted interventions aimed at restoring cardiac homeostasis and improving patient outcomes. To date, there are still no clinical studies directly evaluating alterations of MED subunits in DCM. New discoveries concerning the structure, function, and organization of the MED multiprotein could provide valuable tools for the development of pharmacological molecules which, by binding to specific domains of MEDs, can restore the mechanisms of transcriptional activation/deactivation of disease-specific factors and thus represent a fundamental advance in selective therapeutic approaches. Therefore, future studies are needed to fully understand the pathobiological role of MEDs.

## Figures and Tables

**Figure 1 ijms-25-03057-f001:**
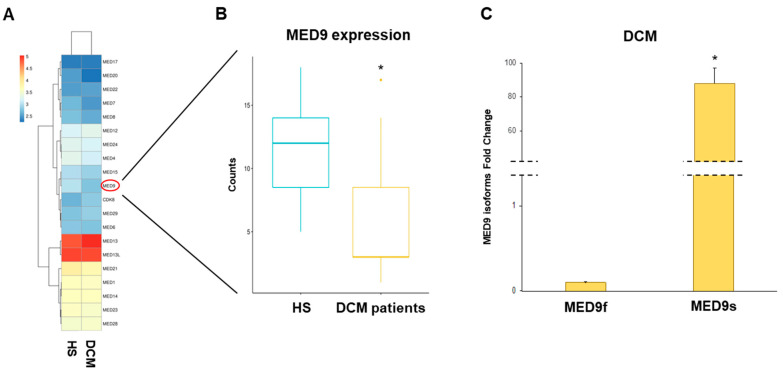
MED subunits alteration into familial DCM patients. (**A**) The heatmap represents MED subunit alteration, combining the MED DEGs into DCM and HS groups. (**B**) MED9 expression levels by RNA-seq. (**C**) Different expression levels of MED9 full (MED9f) and short (MED9s) isoforms. Statistically significant differences between HS and DCM groups were evaluated via the Mann–Whitney U test. * *p* < 0.05 vs. HS. Sample size: n = 22 subjects included, of which n = 11 were healthy subjects (HS), such as organ donors, and n = 11 were DCM patients undergoing heart transplantation.

**Figure 2 ijms-25-03057-f002:**
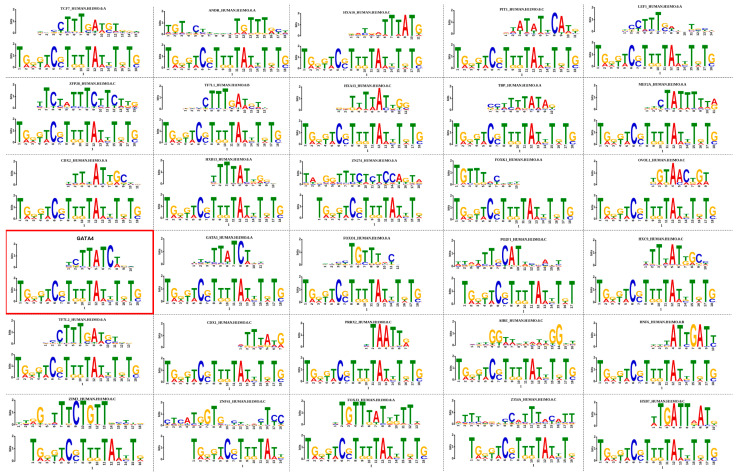
Motif enrichment analysis in familial DCM patients. Motif enrichment analysis identifies candidate TF regulators of tissue-specific DCM development and function. MEME online tool identified more than 30 enriched motifs in DCM group (Appendix A). Among these, GATA4 was identified as a fundamental target in heart development. Sample size: n = 22 subjects included, of which n = 11 were healthy subjects (HS), such as organ donors, and n = 11 were DCM patients undergoing heart transplantation.

**Figure 3 ijms-25-03057-f003:**
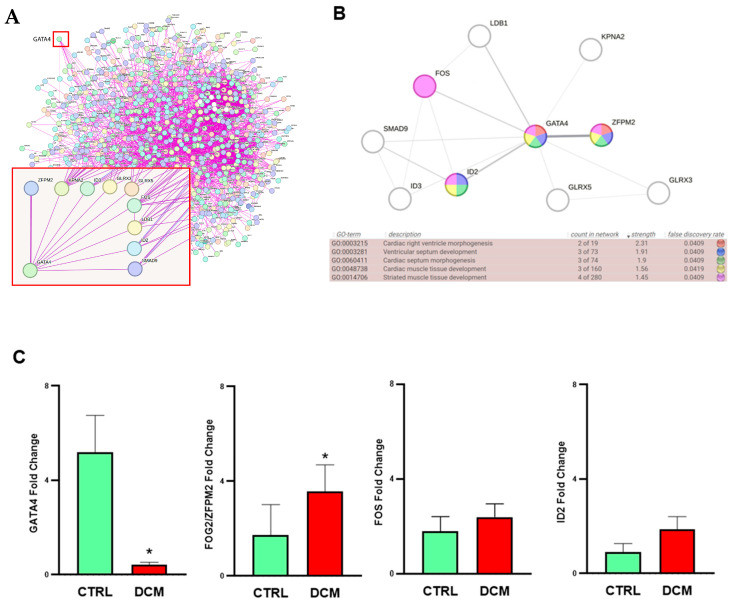
PPI network; GO; STRING analyses and validation data. The figure shows the (**A**) GATA4 PPI network, of which only ZFPM2 showed high-confidence interaction (score > 0.8). The color of nodes indicates the query proteins and the first shell of interactors. The high-confidence interaction for ZFPM2 was indicated with a thicker line. (**B**) GATA4 interactions. GO results were analyzed using STRING tool. (**C**) Quantitative reverse transcriptase PCR (qRT-PCR) was performed to validate data obtained. Bar graph of normalized gene expression show GATA4, FOG2/ZFPM2, FOS, and ID2 analysis in DCM and HS groups. The relative expression was calculated as fold change (FC) between DCM and HS groups and we assigned a value = 1 for the healthy subjects’ (HS) group. Statistically significant difference was evaluated via Mann–Whitney U test for comparison between two groups. * *p* < 0.05 vs. HS. Sample size: n = 22 subjects included, of which n = 11 were healthy subjects (HS), such as organ donors, and n = 11 were DCM patients undergoing heart transplantation.

## Data Availability

Data are contained within the article and Appendix A.

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
