# Peer review of "Familial Dilated Cardiomyopathy: A Novel MED9 Short Isoform Identification"

_ijms, 2024, doi:10.3390/ijms25053057_

Round 1
Reviewer 1 Report
Comments and Suggestions for Authors
The topic of the paper „Familial Dilated Cardiomyopathy: A Novel MED9 Short Isoform Identification“, is interesting and relevant to the field of genetics of heart diseases. The authors have found significant differences in the levels of MED (mediator complex) isoforms between patients with dilated cardiomyopathy (DCM) and healthy subjects and have linked this gene to the development of DCM. This is potentially of great importance for precision medicine. I have only a few minor comments.
Introduction: It should be expanded with two –three sentences describing the existence of autosomal dominant DCM transmission, as well as X-linked, or mitochondrial inheritance pattern.
Mutations in:
titin (DOI: 10.1056/NEJMoa1110186),
laminin (DOI:10.1161/CIRCGENETICS.116.001603.
desmoplakin (DOI: 10.1038/gim.2013.204) and
dystrophin (DOI: 10.1136/hrt.78.6.608) genes deserve to be mentioned.
The aims of the study should be more specific.
M&M:
Under “Study Design and Baseline Patients’characteristics”, authors state that “the subjects were age-matched with a median age of 50 (49.4±16.1) years for the patients and 31 (30.6±13.1) years for heart donor (p<0.01)". They are not matched! Correct this statement.
Figure 1 can be omitted. Simplify the explanation of patients recruitment, i.e. inclusion in the study. I do not see much point of mentioning the 39 initial DCM patients, because the work was done on 11 patients only.
The conclusions are too general; they do not reflect the results of the study. Authors should stick to their findings.
Comments on the Quality of English Language
English language should be slightly improved.
Author Response
Response to REVIEWER 1
The topic of the paper “Familial Dilated Cardiomyopathy: A Novel MED9 Short Isoform Identification”, is interesting and relevant to the field of genetics of heart diseases. The authors have found significant differences in the levels of MED (mediator complex) isoforms between patients with dilated cardiomyopathy (DCM) and healthy subjects and have linked this gene to the development of DCM. This is potentially of great importance for precision medicine. I have only a few minor comments.
01 - Introduction: It should be expanded with two–three sentences describing the existence of autosomal dominant DCM transmission, as well as X-linked, or mitochondrial inheritance pattern.
Thank you for this comment. The Authors improved the text in a useful and more thorough description of the pathology into Introduction section, as suggested.
02 - Introduction: Mutations in: titin (DOI: 10.1056/NEJMoa1110186), laminin (DOI:10.1161/CIRCGENETICS.116.001603. desmoplakin (DOI: 10.1038/gim.2013.204) and dystrophin (DOI: 10.1136/hrt.78.6.608) genes deserve to be mentioned.
In addition to the type of transmission, it is certainly useful to list some mutations characterizing DCM. Thank you for the suggestion.
03 - Introduction: The aims of the study should be more specific.
The Authors re-organized the manuscript, trying to clear the aim of the work.
04 - M&M: Under “Study Design and Baseline Patients’characteristics”, authors state that “the subjects were NOT age-matched with a median age of 50 (49.4±16.1) years for the patients and 31 (30.6±13.1) years for heart donor (p<0.01)". They are not matched! Correct this statement.
Thanks for pointing out this discrepancy. The Authors checked the text and found a typo error.
05 - M&M: Figure 1 can be omitted. Simplify the explanation of patients recruitment, i.e. inclusion in the study. I do not see much point of mentioning the 39 initial DCM patients, because the work was done on 11 patients only.
This suggestion was very helpful. The study was conducted analyzing n=11 DCM patient and n=11 HS. Therefore, the Authors re-elaborated the Methods section, including inclusion criteria, agreeing that mentioning n=39 patients was only confusing. At light of this, they decided to delete Figure 1.
06 - The conclusions are too general; they do not reflect the results of the study. Authors should stick to their findings.
This impression got from reading the conclusions is entirely agreeable. As suggested, the Authors decided to re-write them taking into account the flaws highlighted in the commentary.
07 - Comments on the Quality of English Language: English language should be slightly improved.
We provided English improvement by referring to a native-speaker expert of MDPI journal.

Reviewer 2 Report
Comments and Suggestions for Authors
The authors found MED9 was signficantly downregulated in HF patients. Of the isoforms, MED9s was overexpressed as compared with full length isoform.
The study was comprehensive, from clinical to basic. The study was well done.
I have a simple question, in table 1, in DCM group. there are patients with no HFrEF. Why do the patients belonged to HF group, and why they needed heart transplant.
Comments on the Quality of English LanguageThe English is well written
Author Response
Response to REVIEWER 2
The authors found MED9 was significantly downregulated in HF patients. Of the isoforms, MED9s was overexpressed as compared with full length isoform. The study was comprehensive, from clinical to basic. The study was well done.
01 - I have a simple question, in table 1, in DCM group. there are patients with no HFrEF. Why do the patients belonged to HF group, and why they needed heart transplant.
We thank the Reviewer for the fundamental observation. During the anamnesis, subjects suffering from HF were selected (n=39), but only those presenting with DCM with HFrEF (n=11) underwent organ transplantation and were processed for this study. The HF patients that was not a family history of DCM, so did not undergo heart transplantation. The Authors apologize for the incorrect sentence reported. In light of this, they remodulated the text and deleted the Figure 1.

Reviewer 3 Report
Comments and Suggestions for Authors
Greetings to the authors ! I apologize in advance if my comments seem rather tough, however I do have certain concerns with the manuscript.
Line 30, the MED units are altered in coronary artery disease but this study is about DCM, why mention it randomly here ?
lines 57 to 60, biventricular pacemaker and icd are not reserved for patients "refractory to treatment", they have certain indications and the authors should consult the esc guidelines.
Line 87 the authors mention the early use of cardiac therapy, however clinical therapy is adjusted based on current cardiac parameters and clinical status so what is this preventive therapy the authors are mentioning ?
Lines 93 and 94 authors again mention this so called prophylaxis to genetically determined DCM, but what would this prophylaxis be exactly ? since the authors are quoting clinical implications they should have a more realistic view on the clinical setting of DCM.
Lines 102, 104, the inclusion criteria are quite confusing. Also you compare 39 patients with DCM to 11 healthy patients, why not enroll an equal number of "healthy" patients to serve as a control ? Also, there is no data on the associated pathologies of the so called healthy patients, no mention of other pathologies which can effect the heart such as blood pressure, diabetes, etc.
Again in lines 112 113 it does not appear to be very clear.
lines 291 to 293, in vivo biopsy is rarely performed in real life clinical conditions in patients with DCM, due to the fact that there is little value to this and the diagnosis can be easily done using ultrasound, what are the authors talking about when referring to early differential diagnosis ??
line 349 the authors make bold claims with a "fundamental advance"
In this current early stage, I see this manuscript as unfinished and quite rough around the edges, it needs significant refinement before publication, the methodology is quite confusing, the results invoke clinical results which appear at first hand at least, not as significant as the authors state.
Also I could not access the supplementary files.
Comments on the Quality of English LanguageEnglish needs revising throughout the manuscript.
Author Response
Response to REVIEWER 3
Greetings to the authors! I apologize in advance if my comments seem rather tough, however I do have certain concerns with the manuscript.
01 - Line 30, the MED units are altered in coronary artery disease, but this study is about DCM, why mention it randomly here?
The Authors thank the Reviewer for her/his interesting comments. We have tried to satisfy every suggestion, in order to make the text clearer and more discursive.
The Authors wanted to underline that in the literature, MED subunits have already been found altered in several cardiovascular diseases, including CAD. However, since the sentence can be confusing, the Authors decided to report the results obtained directly in DCM, as suggested.
02 - lines 57 to 60, biventricular pacemaker and icd are not reserved for patients "refractory to treatment", they have certain indications, and the authors should consult the esc guidelines.
As suggested, the Authors modified the text.
03 - Line 87 the authors mention the early use of cardiac therapy, however clinical therapy is adjusted based on current cardiac parameters and clinical status so what is this preventive therapy the authors are mentioning?
As suggested, the Authors modified the text to clarify the hypothesis of our study.
04 - Lines 93 and 94 authors again mention this so called prophylaxis to genetically determined DCM, but what would this prophylaxis be exactly? since the authors are quoting clinical implications they should have a more realistic view on the clinical setting of DCM.
To clarify the concept, the Authors rephrased the sentence, avoiding focusing on a hypothetical prophylaxis, but rather identifying the underlying pathophysiology that could guide future treatments aimed at repairing or replacing the existing genetic mutation, or targeting specific epigenetic and/or inflammatory factors altered in DCM.
05 - Lines 102, 104, the inclusion criteria are quite confusing. Also you compare 39 patients with DCM to 11 healthy patients, why not enroll an equal number of "healthy" patients to serve as a control? Also, there is no data on the associated pathologies of the so-called healthy patients, no mention of other pathologies which can affect the heart such as blood pressure, diabetes, etc.
The Authors reported the inclusion criteria more in detail. Specifically, the study included n=50 subjects, of which n=11 were organ donors, who died due to accidental death and used as healthy controls. HF patients (n=39) were divided into two groups: DCM (n=11) and no DCM (n=29). Among the 2 groups, only the DCM group, reporting a reduced ejection fraction (HFrEF), which underwent heart transplantation, was processed by NGS. This group included patients with determinate familial diagnosis.
Since the manuscript was confusing in the Methods section, the Authors rephrased for clarity. Specifically, the number of myocardial tissue samples from healthy subjects and DCM patients is the same (n=11 HS and n=11 DCM). Moreover, since also Figure 1 was confusing, they decided to remove it. Based on the available data registered by clinicians at the time of death, it emerged that none of the 11 healthy subjects had significant cardiac-metabolic pathologies. About 50% showed overweight (BMI between 25 to 28) and a history of smoking habits.
06 - Again in lines 112 113 it does not appear to be very clear.
As suggested, the Authors rewrote the text.
07 - lines 291 to 293, in vivo biopsy is rarely performed in real life clinical conditions in patients with DCM, due to the fact that there is little value to this and the diagnosis can be easily done using ultrasound, what are the authors talking about when referring to early differential diagnosis??
The Authors are aware that in vivo biopsy is not necessary for DCM diagnosis. Nonetheless, they performed this study with the aim of identifying specific biomolecular targets, which could be altered during DCM compared to the healthy state.
As previously reported, the Authors rephrased the entire text, keeping in mind the interesting results achieved.
08 - line 349 the authors make bold claims with a "fundamental advance"
See #07 comment.
09 - In this current early stage, I see this manuscript as unfinished and quite rough around the edges, it needs significant refinement before publication, the methodology is quite confusing, and the results invoke clinical results which appear at first hand at least, not as significant as the authors state.
The Authors have edited all the suggested comments, with the aim of making the manuscript clearer and closer to the real molecular and cellular alterations of a heart in DCM compared to a healthy heart. The identification of a new isoform of an important factor such as MED is for the authors an interesting advancement in understanding the onset of the pathology. They therefore hope that the changes made in the different paragraphs (Methods, Results, Discussion sections) are as close to the idea that the Reviewer envisaged.
10 - Also I could not access the supplementary files.
This greatly displeased the authors. Supplementary data will be re-uploaded during the review submission phase.
11 - Comments on the Quality of English Language: English needs revising throughout the manuscript.
We provided English improvement by referring to a native-speaker expert of MDPI journal.
